

# A simple equation for the surface-elevation feedback of ice sheets

A. Levermann[1,2,3*] & R. Winkelmann[1,3]

[1]*Potsdam Institute for Climate Impact Research, Potsdam, Germany.*

[2]*LDEO, Columbia University, NY, USA.*

[3]*Institute of Physics, Potsdam University, Potsdam, Germany.*

*Correspondence to: anders.levermann@pik-potsdam.de*





**Abstract**:

In recent decades, the Greenland Ice Sheet is been losing mass and thereby contributed to global sea-level rise. The ice loss is likely to increase under future warming. Beyond a critical temperature threshold, a meltdown of the Greenland Ice Sheet is induced by the self-enforcing feedback between its lowering surface elevation and its increasing surface mass loss: The more ice is lost, the lower the ice surface reaches into the atmosphere and the warmer the surrounding air becomes which fosters melting and further ice loss. The rate of ice loss is highly relevant for coastal protection worldwide. The computation of this rate so far relies on complex numerical models as it should be. In order to contribute a little to the conceptual understanding, we derive here a simple equation for the self-enforcing feedback and use it to estimate the melt time for different levels of warming using three observable characteristics of the ice sheet itself and its surroundings. When the volume loss is dominated by the feedback, the resulting logarithmic equation unifies existing numerical simulations and shows that the melt time depends critically on the level of warming with a critical slowing-down near the threshold: The median time to lose 10% of the present-day ice volume varies between about 3500 years for a temperature level of 0.5°C above the threshold and 500 years for 5°C. Unless future observations show a significantly higher melting sensitivity than currently observed, a complete melt down is unlikely within the next 2000 years without significant ice-dynamical contributions.





## 1. Introduction

Anthropogenic climate warming by expanding ocean waters and melting ice is raising global sea level (IPCC, 2013). Over the two past decades, the Greenland Ice Sheet has lost mass at an accelerating pace (Bamber et al., 2000; Box et al., 2012; van den Broeke et al., 2009; Fettweis et al., 2013; Mernild et al., 2011; Nick et al., 2009; Rignot et al., 2008, 2011; Shepherd and Wingham, 2007; Thomas et al., 2011). The ice loss is likely to increase under unabated greenhouse-gas emissions (Clark et al., 2016; Fettweis et al., 2013; Goelzer et al., 2012; Graversen et al., 2011; Harper et al., 2012; Huybrechts et al., 2011; Levermann et al., 2013; Nowicki et al., 2013; Price et al., 2011). Numerical simulations suggest that a decline of the Greenland Ice Sheet is inevitable once its surface temperature permanently exceeds a certain threshold (Charbit et al., 2008; Greve, n.d.; Huybrechts and Wolde, 1999; Huybrechts et al., 2011; Ridley et al., 2005, 2010; Robinson et al., 2012; Solgaard and Langen, 2012). If and when this temperature threshold is passed, depends critically on past and future greenhouse-gas emissions (Fettweis et al., 2013; Goelzer et al., 2013; Gregory et al., 2004a; Rae et al., 2012). Even if emissions were reduced to zero, temperatures would not drop significantly for thousands of years because of the long life-time of anthropogenic $CO_2$ in the atmosphere and reduced oceanic heat uptake if oceanic convection is extenuated (Allen et al., 2009; Solomon et al., 2009; Zickfeld et al., 2013). This implies a possible commitment of a melt-down of the Greenland Ice Sheet in the near future which would eventually raise global sea-level by more than 7 meters (Howat et al., 2014b). Whether this occurs on a multi-centennial or rather a multi-millennial time scale is of relevance for coastal planning. The framework that we provide here can also be used to imclude new physical processes that might be discovered in the future, e.g. potential changes in surface albedo through melting (Box et al., 2012) or aerosol-induced surface melt or the lack thereof (Polashenski et al., 2015).

Here we first recap the Vialov profile and add a simple representation of the surface-elevation feedback towards a governing equation for a steady-state ice-sheet in zero dimension (section 1), then we derive the critical warming threshold for the existence of an ice sheet in this simple model (section 2). In section 3 we derive a simple time-evolution equation for the decay of the ice sheet after surface temperatures have exceeded the threshold. Finally we use observational estimates of the three parameters that enter the model to estimate the decay time of the ice sheet



56    under melting above the threshold. Here solid ice discharge is neglected as well as any other ice

57    sheet dynamics (Andresen et al., 2012; Howat and Eddy, 2012; Moon et al., 2012; Nick et al.,

58    2009; Price et al., 2011; Straneo et al., 2011; Walsh et al., 2012).



## 2. Governing equation for shallow-ice steady states under surface-elevation feedback

A nonlinear threshold behavior is generally associated with a fundamental self-enforcing feedback and thereby an associated system memory e.g. (Levermann et al., 2012). For the Greenland Ice Sheet, such a feedback is given by the interaction between surface elevation and surface melting (Weertman, 1961). For illustration, we include this feedback in a well-established highly idealized ice-profile of an ice-sheet in one dimension, the so-called Vialov-profile (Vialov, 1958). We introduce the surface-elevation feedback in the simplest possible way by assuming that the surface melt rate depends linearly on the surface temperature and that the temperature decreases linearly with the height of the ice surface following a constant atmospheric lapse rate.

We consider a highly simplified flowline model for an isothermal ice sheet grounded on a flat and rigid bed. The solution of the shallow-ice approximation in one dimension for the ice-sheet elevation under these simplifying assumptions is called the Vialov-profile:

$$\tilde{h}(x) = h_m \left( 1 - (x/L)^{(n+1)/n} \right)^{n/(2n+2)} \tag{1}$$

where $h_m$ is the maximum surface elevation and $n$ is Glen's flow law exponent (Glen, 1955). The inherent assumption of isothermal ice is a strong simplification, but we do not aim for a realistic representation of the ice flow but will derive a measure for the average height of the ice sheet and its dependence on changes in the surface mass balance. The surface mass balance is considered to be homogeneous at a value, $a$, which will later be considered dependent on the surface elevation. The overall horizontal extension of the ice sheet is set to $L$, and it is thereby assumed that any ice flow across this point is calved off into ice bergs. This situation represents a confined ice-bearing bedrock topography as in most of Greenland's interior (Howat et al., 2014a).

The mean surface elevation can then be computed to be

$$\bar{h} = L^{-1} \int_0^L dx\, h(x) = \omega \cdot h_m \tag{2}$$

It is proportional to the maximum surface elevation $h_m$ with a proportionality factor

$$\omega \equiv \int_0^1 d\xi \left( 1 - \xi^{(n+1)/n} \right)^{n/(2n+2)} \tag{3}$$



which only depends on the flow law exponent.
The maximum surface elevation is determined by the surface mass balance $\tilde{a}$ and the ice softness
$\tilde{A}$
$h_m = 2^{(n-1)/(2n+2)} \cdot L^{1/2} \cdot \left( \frac{(n+2)\tilde{a}}{(\rho g)^n \tilde{A}} \right)^{1/(2n+2)}$    (4)
with $\rho$ being the ice density and $g$ the gravity constant. We normalize all three quantities by
defining $h \equiv \omega \cdot h_m / \overline{h_0}$, $a \equiv \tilde{a} / a_0$ and $A \equiv \tilde{A} / A_0$ where $a_0$ is the accumulation rate on the
ground, i.e., in the absence of an ice-sheet, and $A_0 = a_0 / \left( (\rho g)^n (\varepsilon \cdot L)^{(n+1)} \right)$ with $\varepsilon = H/L$ being
the typical height-to-width ratio. $\overline{h_0}$ is the equilibrium line of the considered ice sheet in the
initial equilibrium situation. Values for $a_0$, $\overline{h_0}$ and $L$ are later chosen to resemble the conditions
of the Greenland Ice Sheet.
The non-dimensional surface elevation, h, of the ice sheet can then be expressed as
$h = \left( \frac{a}{A} \right)^{1/m}$    (5)
For the Vialov profile m=2(n+1) where n denotes the Glen flow-law exponent observed to be
around n=3 which yields m=8. We introduce the surface elevation feedback in its simplest form
through a dependency of the surface melt rate on the surface elevation:
$a = a_0 + \gamma \Gamma \cdot h$    (6)
with the atmospheric lapse rate $\Gamma > 0$. $\gamma$ denotes the melting sensitivity of the ice surface, i.e. the
increase in surface melt-rate per degree of warming, which is regularly measured and comprises
a large number of physical processes (e.g. (Box, 2013)). For simplicity we rescale the surface
mass balance by the constant ice softness parameter, $A$, which is considered to be constant. The
steady state solution for the surface elevation of the ice-sheet is thus governed by the following
equation
$h^m - \gamma \Gamma \cdot h - a_0 = 0$    (7)
which has two positive solutions for $h$ as long as the surface mass balance on the ground is
negative, i.e., $a_0 < 0$. Note that the surface mass balance can be positive even if $a_0 < 0$. If the



ice-sheet is in an unstable configuration, a slight perturbation will either cause it to converge into
the stable state with a positive surface mass balance or to melt-down completely.
Our simple approach qualitatively captures the basic hysteresis behavior of the Greenland Ice
Sheet caused by the surface-elevation feedback (Fig. 1): For a given surface temperature a stable
state of the ice sheet (red line) annihilates an external perturbation in surface elevation by
changes in surface mass balance (grey arrows). The unstable solution branch defines the basin of
attraction for the stable state. A surface elevation that is lower than the unstable solution branch
cannot be sustained. In that case the melting reduces the surface elevation to practically zero
even without further external perturbation (grey arrows). Beyond a certain surface temperature
threshold (vertical dotted line) no ice sheet can be sustained.

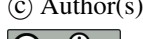



### 3. Critical surface mass balance in steady state

As illustrated in Fig. 1, there is a critical temperature above which the ice-sheet is not sustainable. Let us denote the corresponding surface elevation by $h_c$. The critical point $(T_c, h_c)$ has to fulfill two conditions. First, it has to be a solution of the governing equation and second it has to be a minimum of the function

$$F(h) = h^m - \gamma \, \Gamma \cdot h - a_0 \tag{8}$$

which we can determine by setting the derivative of F to zero.

Consequently,

$$h_c = \left(\frac{\Gamma \cdot \gamma}{m}\right)^{1/(m-1)} \tag{9}.$$

Inserting this into the governing equation yields the critical surface mass balance at the ground

$$a_{0c} = -(m-1) \cdot \left(\frac{\Gamma \cdot \gamma}{m}\right)^{m/(m-1)} \tag{10}.$$

For illustrative purposes we have assumed $a_0$ to decline linearly with the surrounding temperature and plotted the solution of equation (7) against that temperature with an arbitrary off-set in Fig. 1.



## 4. A simple temporal equation for the surface-elevation feedback

Based on the governing equation, we can derive the critical surface mass balance and surface elevation below which a meltdown of the ice-sheet is inevitable. Let us define the time $\tau_\alpha$ as the time it takes to melt a fraction $\alpha$ of the initial ice volume and the threshold temperature $T_c$ as the temperature above the pre-industrial level at which the surface mass balance becomes negative. Robinson et al. (2012) find a range of 0.8 – 3.2°C for the threshold warming beyond which no ice sheet can be sustained on Greenland. Their best estimate for the threshold is 1.6°C above pre-industrial level. Ridley et al. (2010) find that in their model the ice sheet cannot be sustained for a warming of 2°C. Some studies assume that the threshold is associated with a mean negative surface mass balance (Gregory et al., 2004b; Ridley et al., 2005; Toniazzo et al., 2004). In Fig. 2 we use 1.6°C as a threshold value for both models. This number can be easily adjusted if new estimates are obtained.

For a fixed anomalous melt rate $\Delta a_0 = -\gamma \cdot \Delta T$ in response to an anomalous temperature increase $\Delta T$ above this threshold temperature, the decay time without any feedbacks would be

$$\tau_0 = -\frac{h_0}{\Delta a_0} = \frac{h_0}{\gamma \cdot \Delta T} \tag{11}$$

Since the surface temperature increases with decreasing elevation, this zero-order estimate for the decay time is higher than the actual value. As a first-order correction to the situation of fixed melting, let us assume that the anomalous surface mass balance behaves as

$$\Delta a = \Delta a_0 + \frac{1}{\tau_\gamma} \cdot (h - h_0) \tag{12}$$

where $\tau_\gamma = 1/(\gamma \cdot \Gamma)$.

From the relation $dh/dt = \Delta a$, we then obtain

$$\frac{d\Delta h}{dt} = -\Delta a_0 + \frac{\Delta h}{\tau_\gamma}, \tag{13}$$

if $\Delta h \equiv h_0 - h$ is defined as the reduction in height. For a time-dependent melting induced by surface warming $\Delta a_0 = \gamma \cdot \Delta T$ the general solution of equation (13) is





$$\Delta h(t) = \gamma \cdot \int_0^t dt' \, \Delta T(t') \cdot e^{(t-t')/\tau_\gamma} \qquad (14)$$
This equation corresponds to a linear response theory with the melting $\gamma \cdot \Delta T$ as forcing and an
exponential response function
$$R(t') = e^{t'/\tau_\gamma} \qquad (15)$$
Linear response theory states that the convolution of equation (14) yields the linear response of
the system (Good et al., 2011; Winkelmann and Levermann, 2013). Note that generally linear
response theory is used as an approximation of a non-linear system to relatively weak forcing. In
these circumstances the response function has to decline with time because it represents the
history of the system's response to past perturbation. For example, if the response function was
a declining exponential $R(t') = e^{-t'}$ that would mean that the effect of forcing that occurred in
the past, i.e. prior to the time t that is considered, becomes exponentially less relevant for the
current system response. Here however the response function is increasing with time which
means that the past deviation from the steady state is amplified which is exactly what an unstable
situation should do. The exponent $1/\tau_\gamma$ can be considered the Lyaponov exponent of the system.
Given the boundary condition $\Delta h(t=0)=0$, for a constant temperature increase $\Delta T$, equation (14)
becomes
$$\Delta h(t) = h_0 \cdot \left( \frac{\tau_\gamma}{\tau_0} - \frac{\tau_\gamma}{\tau_0} \cdot e^{t/\tau_\gamma} \right) - \frac{h_0}{\tau_0} - \frac{h_0}{\tau_\gamma} \qquad (16).$$
The decay time for a relative volume reduction of $\alpha$ is then given by:
$$\tau_\alpha = \frac{1}{\gamma \Gamma} \cdot log\left( 1 + \alpha \cdot \frac{\Gamma \cdot h_0}{\Delta T} \right) \qquad (17),$$
where $log$ denotes the natural logarithm. Equation (17) is denoted the *decay-time equation*
hereafter.



## 5. Estimating the Melt Time of the Greenland Ice Sheet from Observables

In this simplified approach, the collapse time is thus a function of three observable quantities: the equilibrium-line altitude, $h_0$, the atmospheric lapse rate, $\Gamma$, and the melting sensitivity to temperature, $\gamma$. The average equilibrium-line altitude of the Greenland Ice Sheet is at about 1150 meters (Box & Steffen 2001)). The observed range for the atmospheric lapse rate is estimated to be between $5 \pm 2$ °C/km (Fausto et al. 2009; Gardner & Sharp 2009), and current estimates for the melting sensitivity scatter around $4.4 \pm 2$ cm/year/°C (Box 2013). In order to obtain an estimate of the decay time and the uncertainty around this estimate we use equation (17) and chose the lapse rate and melting sensitivity uniformly randomly from these observed intervals (Tab. 1, Figs. 2 – 4).

Existing numerical simulations for a decay of the Greenland Ice Sheet (Ridley et al. 2010; Robinson et al. 2012) differ in their trajectories for the total ice volume, but exhibit a characteristic functional form when the relative ice volume is expressed as a function of the temperature anomaly above the critical temperature threshold (Fig. 2). This characteristic relation is captured by our first-order equation for the decay time, embedding the results from process-based models into a simple analytical framework. This approach provides a good approximation if, on the one hand, the volume loss is significantly large for the surface-elevation feedback to become relevant and, on the other hand, the melting is dominating the ice loss compared to the dynamic ice discharge.

Following the decay-time equation (17), the observational constraints for the atmospheric lapse rate, $\Gamma$, and the melting sensitivity, $\gamma$, translates into an uncertainty range for the melt time of the Greenland Ice Sheet, assuming uniform probability distributions for both $\Gamma$ and $\gamma$ within the above intervals. Fig. 2 shows the histograms of the time until 10% of its present-day ice volume (corresponding to 0.7 m global sea-level rise) are melted for different warming scenarios. The melt time depends strongly on the level of warming beyond the temperature threshold: The median estimate varies from more than 2000 years for a warming of +1°C to less than 500 years for a warming of +5°C.

Since the melt time is a monotonically decreasing function of both the lapse rate and the melting sensitivity, the upper and lower limits of the estimates can be directly computed from the



observed uncertainty interval of these quantities. However, the functional form of equation (17)
introduces a specific structure into the histogram of the melt time which is highly skewed
towards the low end (Tab. 1 and Fig. 4).
The simple equation provided here is clearly limited in its applicability. Since it does not account
for any dynamic discharge or even ice motion the results from equation (17) strongly deviate
from numerical simulations when the ice has time to adjust dynamically to the volume loss. This
can be seen for a stronger ice loss of 50% of the initial volume (Fig. 3). Also the role of the ice
material properties is comprised into one parameter, the melting sensitivity of the ice to a
temperature increase at the surface. This sensitivity will in general vary not only with time but
also spatially and due to the melting itself. Similarly the feedback role of the surrounding climate
is represented by only one parameter, the atmospheric lapse rate which will again vary spatially
but also with time as the ice surface declines.
The dynamic discharge from Greenland is strongly limited by the ice sheet's bottom topography,
for which estimates yield an upper bound of approximately 5-13 cm during the next century
(Graversen et al., 2010; Price et al., 2011). Over a period during which the ice loss is dominated
by the feedback and the ice-dynamic effect is limited, our approach provides a quantitative
estimate of the melt time based on observable quantities. Equation (17) can thus be used when
new observations suggest an altered melting sensitivity or changes in the atmospheric response
to Greenland ice loss.



## 6. Discussion and conclusion

Our estimate for the decay time captures the characteristic slow-down near the critical threshold as can be seen from the divergence of the decay time, $\tau_\alpha$, in the limit of vanishing warming above the threshold (equation (17)). The simple equation of the decay time quantitatively reproduces the range given by simulations with process-based models. The feedback becomes more dominant near the threshold compared to larger temperature increase for which the external climatic forcing is more relevant (Fig. 5). For these curves in this figure we used the central values of the parameters, i.e. equilibrium-line altitude $h_0$=1150m, atmospheric lapse rate $\Gamma$=5 °C/km and melting sensitivity $\gamma$=4.4 cm/year/°C.

For a temperature increase of 5°C, which could be reached within this century (IPCC, 2013), the median rate of sea-level contribution is about 1.4 mm per year which is about four times that of its current contribution (Rignot et al., 2011). Even for extremely high temperatures however, the Greenland Ice Sheet cannot melt infinitely fast – our results show that a complete disintegration within the next two millennia is highly unlikely unless ice dynamics effects become dominate or the melting sensitivity is significantly higher than currently observed. For a global mean temperature increase below two degrees, as agreed upon during the 1015 Paris UNFCCC climate summit, the threshold temperature would only be exceeded mildly and the decay time of the Greenland ice sheet would be mult-millennial.



| Volume loss | | 0.5℃ | 1℃ | 2℃ | 3℃ | 4℃ | 5℃ |
|---|---|---|---|---|---|---|---|
| 10% | Lower | 2140 yr | 1320 yr | 760 yr | 530 yr | 410 yr | 330 yr |
| | Median | 3430 yr | 2040 yr | 1140 yr | 790 yr | 610 yr | 500 yr |
| | Upper | 7290 yr | 4120 yr | 2210 yr | 1520 yr | 1150 yr | 930 yr |
| 50% | Lower | 4920 yr | 3600 yr | 2460 yr | 1900 yr | 1550 yr | 1320 yr |
| | Median | 8740 yr | 6170 yr | 4040 yr | 3040 yr | 2450 yr | 2090 yr |
| | Upper | 20740 yr | 13920 yr | 8640 yr | 6310 yr | 4980 yr | 4120 yr |
| 100% | Lower | 6340 yr | 4920 yr | 3600 yr | 2910 yr | 2460 yr | 2140 yr |
| | Median | 11610 yr | 8730 yr | 6160 yr | 4840 yr | 4020 yr | 3500 yr |
| | Upper | 28710 yr | 20740 yr | 13920 yr | 10630 yr | 8640 yr | 7290 yr |

**Table 1**: **Decay time.** Time period after which different percentages of volume loss have occurred at different warming levels. Provided are the median values of the distributions from Figures 2 and 3 together with the lower and upper limit that are derived respectively from the upper and lower limits of the uncertainty range of the observed melting sensitivity and atmospheric lapse rate.



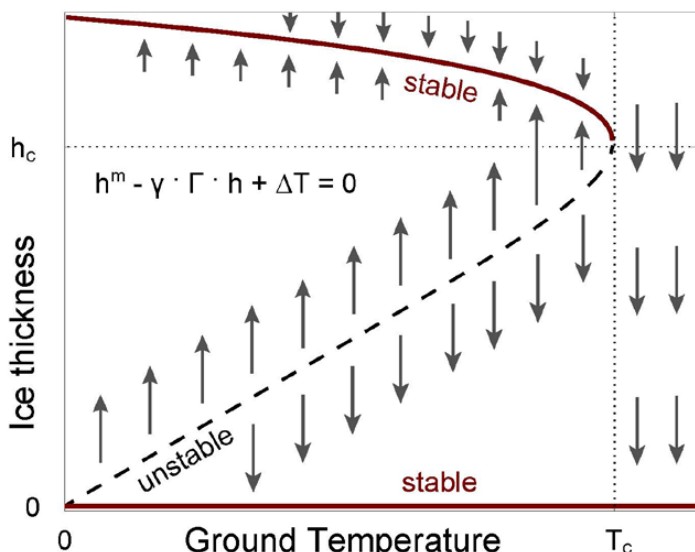

251

**Figure 1: Ice-sheet hysteresis.** *If the ice-sheet is in an unstable configuration (dashed black branch), a slight perturbation will either cause it to converge into the stable state (upper red branch) or to melt-down completely. For a given temperature, the dotted line gives the critical surface elevation (section 3). If the surface elevation is lower than $h_c$, a complete meltdown of the ice sheet is inevitable. Once the temperature threshold, $T_c$, is crossed, the time for a collapse of a certain fraction of the ice-sheet can be estimated via equation (17).*





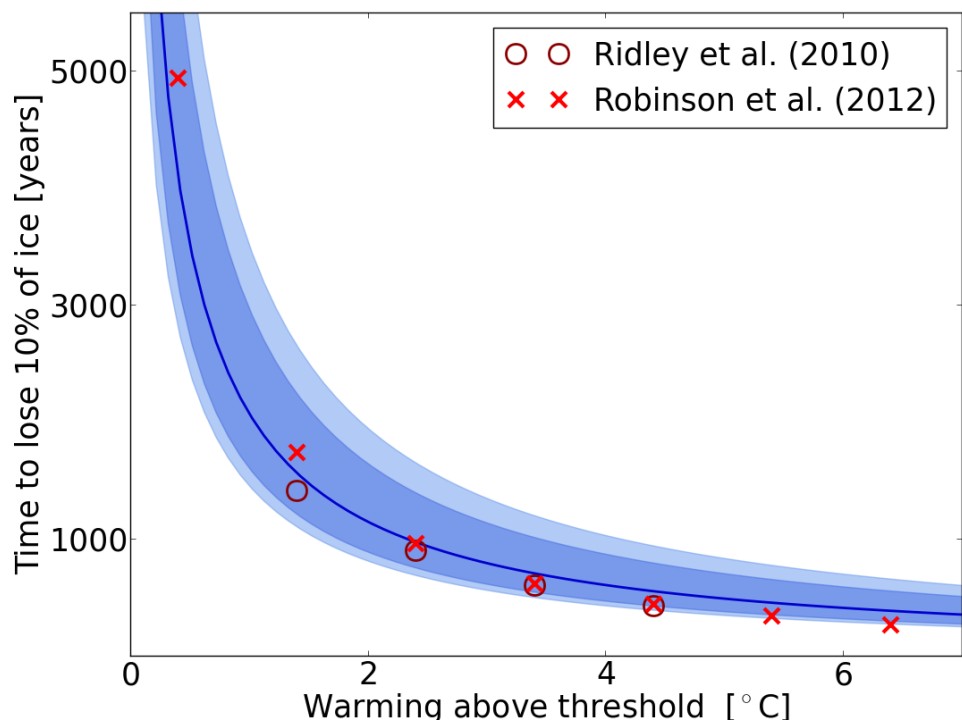

258

*Figure 2*. ***Decay-time of the Greenland Ice Sheet.*** *The decay time depends critically on the level*

*of warming above the temperature threshold. Shown are the median (black line) and the likely*

*(18% to 83% quantiles, dark blue shading) and very likely (5% to 95% quantiles, light blue*

*shading) ranges for the time to melt 10% of the present-day ice volume, estimated via equation*

*(17). The red circles and crosses indicate the results from process-based model simulations by*

*Ridley et al. (2010) and Robinson et al. (2012), respectively.*



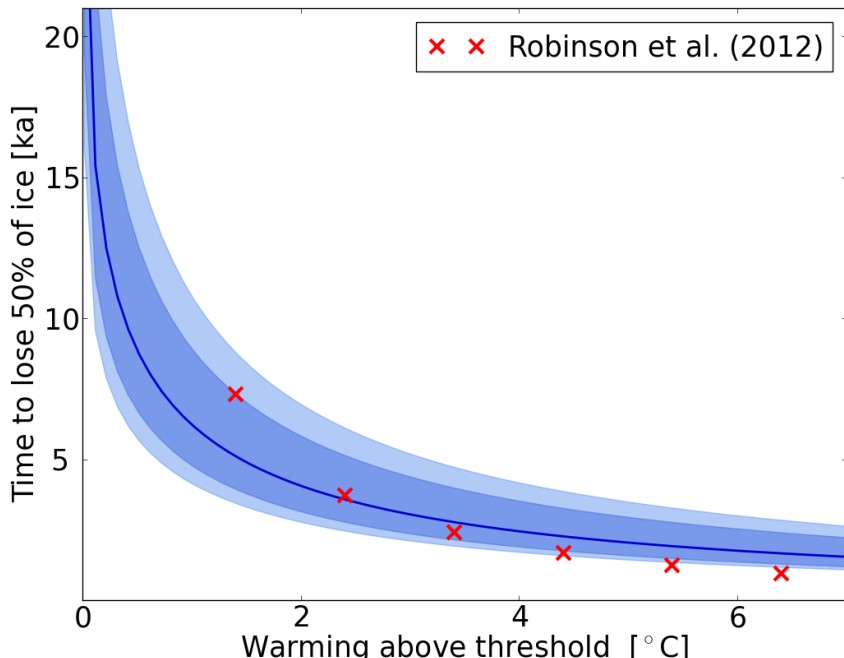

265

**Figure 3: Time until 50% of the Greenland Ice Sheet are melted.** *Shown are the median (black*

*line) and the likely (18% to 83% percentiles, dark blue shading) and very likely (5% to 95%*

*percentiles, light blue shading) ranges for the time to melt 50% of the present-day ice volume,*

*estimated via the equation for the decay time* $\tau_\alpha$. *The red crosses indicate the results from*

*process-based model simulations by Robinson et al. (2012).*



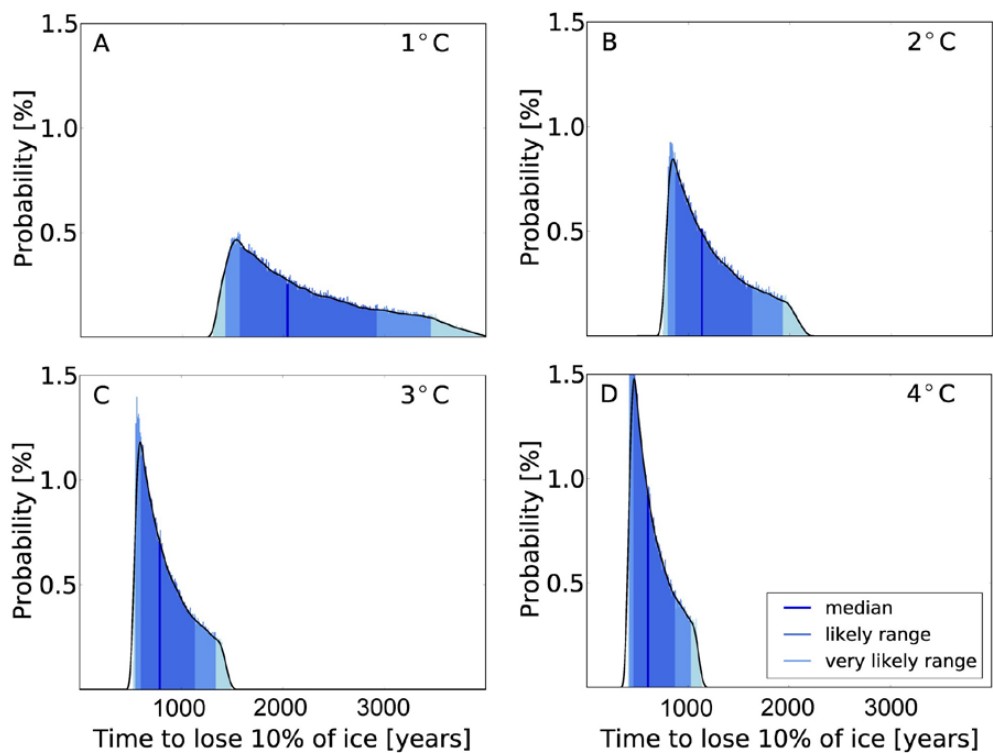

**Figure 4. Likelihood for 10%-decay of Greenland Ice Sheet.** *Shown are the probabilities for the ice-sheet to lose 10% of its initial ice volume in a certain time period for surface warming of +1°C (A), +2°C (B), +3°C (C) and +4°C (D) above the threshold. The median is indicated by the black line, and the likely and very likely ranges are shaded in dark and light blue, respectively.*





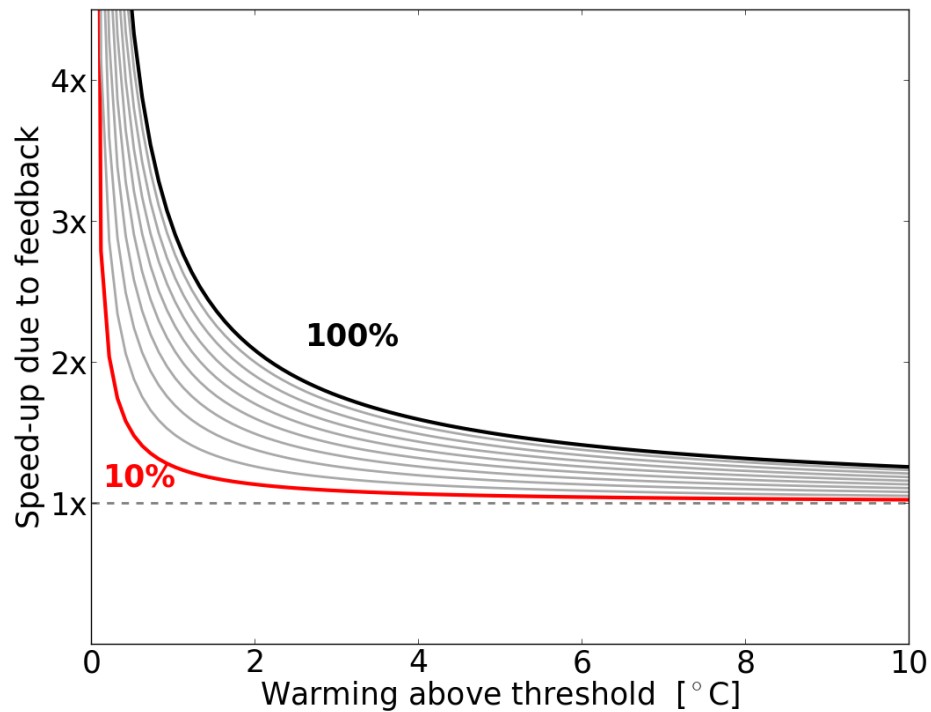

278

*Figure 5: Role of surface elevation feedback in melting of Greenland ice sheet declines with*

*increasing temperature. Shown is the ratio of melting time with surface-elevation feedback over*

*melting time without the feedback $\tau_\alpha/\tau_0$. Each line represents the ratio for a loss of different*

*percent of the initial ice volume. The red line shows the ratio of the decay time with feedback*

*over the decay time without feedback for a 10% ice loss (corresponding to figures 2 and 4). The*

*influence of the feedback becomes less dominant with stronger warming above the critical*

*threshold (x-axis). Near the threshold the melting time without feedback diverges stronger*

*(1/ΔT) than the melt time with feedback which declines logarithmically.*





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
