# Peer review of "A simple equation for the melt-elevation feedback of ice"

_The Cryosphere, 2016_

## Short Comment (SC1) · 12 Apr 2016

Hi there,

Interesting paper!

In reading it I was struck by similarities to a classic modeling study of Oerlemans et al., 1981, particularly Section 4:

"Some basic experiments with a vertically-integrated ice sheet model", Tellus, 33, pg. 1-11.

For example, compare your characteristic slow-down near the critical threshold to Oerleman's 'intransitive' zone, or your Figure 1 to Oerleman's Figure 7.

At the least I would suggest adding this early work as a reference, perhaps along with

discussion of differences and similarities between your results and Oerleman's if you feel they are similar enough to justify such a contrast.

---

## Author Comment (AC1) · 12 Apr 2016

Dear Jeremy,

thank you very much for this hint. We were indeed not aware of the paper by Oerlemans. It is very interesting and relevant and we will definitely discuss it in our revisions.

Best wishes, Anders & Ricarda
* * *

---

## Referee Comment (RC1) · Anonymous Referee #1 · 2 May 2016

**General Comments**

This paper shows an interesting way of deriving the melt time of the Greenland Ice Sheet for different warming levels using a very simple approach based on three observable quantities: the equilibrium-line altitude (ELA), the atmospheric lapse rate and the melting sensitivity of the ice surface to temperature. The most interesting result is that the derived decay time quantitatively reproduces the range given by existing process-based numerical simulations. This study is relevant in the current context of Greenland Ice Sheet mass loss.

However, the approach suffers from several drawbacks that we detail in the Specific Comments below, especially the non-applicability of the decay time equation if dynamic discharge is taken into account, the lack of experiments to confirm the results given by the proposed equation, the lack of connection between different sections of the paper and the poor discussion. Therefore, we advise the authors to revise the paper either for providing a more substantial analysis of their work or for summarizing their results into a brief communication.

**Specific Comments**

1. The decay time equation proposed here does not take into account ice dynamics, as the authors state in section 5. However, a number of studies have shown that, even if the contribution of the dynamic part in Greenland ice loss seems to be less important than surface mass balance (SMB) changes, it is still quite substantial. One of the most recent modeling studies about this topic (Furst et al., 2015, TC) shows that 40

2. Even if we assume that ice loss only comes from SMB changes (which is the case of this study), the study lacks some proofs that the decay time equation is robust against process-based studies. Only Figure 2 clearly shows that the results agree well with two process-based numerical simulations, even if it does not show the time to loose 10

3. It is not straightforward to understand how sections 2 and 3 really fit into the paper since the authors do not use the equations (1) to (10) related to the Vialov profile and the critical SMB for deriving the decay time equation (17), except equation (6) that relates surface melt rate and elevation. It is nice to see how the critical SMB and surface elevation below which a meltdown is inevitable are calculated but they are not really used in computing the main results of the paper (since the decay time only depends on the warming level, lapse rate, ELA and melting sensitivity). As far as we understood, one of the main purposes of sections 2 and 3 is to show where Figure 1 (which is quite nice) comes from.

4. The discussion clearly misses a robust analysis of the results. For example, some drawbacks related to the use of the decay time equation are presented at the end of section 5 but we think that they should really go into the discussion and be more detailed.

5. The whole paper talks about the surface-elevation feedback but in reality this is the SMB-elevation feedback (IPCC, 2013; Edwards et al., 2014, TC; Goelzer et al., 2013, J. Glaciology). Furthermore, the paper does not talk about the feedback of ice sheets in general but of the Greenland Ice Sheet in particular. Finally, the results of the paper

focus less on the SMB-elevation feedback than on the melt time. Therefore, we suggest a different title: 'A simple equation for the melt time of the Greenland Ice Sheet'.

6. In section 1 (Introduction), the first paragraph is very long and could be separated into two different paragraphs, one with the general Greenland ice loss context and the other one with the temperature threshold and the SMB-elevation feedback. In any case, the link with the last paragraph of section 1 is not really done. We would add a clear explanation about the SMB-elevation feedback and the importance of determining the melt time for Greenland.

[Figure]

**Technical Corrections**

P2, L9: 'has been loosing' instead of 'is been loosing.

P2, L12: 'the' instead of 'The'.

P2, L13: Rephrase 'the lower the ice surface reaches into the atmosphere' since this is not clear.

P2, L14-15: The sentence 'The rate of ice loss is highly relevant for coastal protection worldwide' does not really fit here. It could go in the beginning or at the end of the abstract.

P2, L16: Delete 'as it should be'.

P2, L16: Is the bit 'In order to contribute a little to the conceptual understanding' really needed? We would remove it.

P2, L18: We would cite the three observable 'characteristics', which we think are better defined as 'parameters'.

P2, L20: 'critically depends' instead of 'depends critically'.

P2, L21: Use of 'critical' and 'critically' in the same sentence. Maybe replace 'critically' by 'strongly'.

P2, L21: 'the' instead of 'The'.

P2, L24: 'meltdown' instead of 'melt down'.

P3, L27: The first sentence is not totally accurate. Maybe: 'Global sea level rise has been raising in the past decades mainly due to ocean thermal expansion and melting ice (Church et al., 2013).' The last reference is more accurate than 'IPCC (2013)'.

P3, L28: 'past two decades' instead of 'two past decades'.

P3, L29-31: We think that some older references could be deleted and some newer

studies could be added, e.g. Kjeldsen et al. (2015, Nature) who study the Greenland ice loss since 1900 using aerial imagery, Khan et al. (2015, Reports on Progress in Physics) who provide a review of Greenland Ice Sheet mass balance, Shepherd et al. (2012) who provide results from the Ice sheet Mass Balance Inter-comparison Exercise (IMBIE).

P3, L36: 'Greve, 2000' instead of 'Greve, n.d.'.

P3, L38: 'critically depends' instead of 'depends critically'.

P3, L43: The authors need to agree whether they use 'meltdown' or 'melt-down' throughout the article (see also L24).

P3, L45: We did not find that Howat et al. (2014) mention a sea level rise contribution from Greenland of 7 m. Maybe Gregory et al. (2004, Nature Brief Communications) is a more suitable reference. Please also check your references for Howat et al. (2014) because you list both the TC and TCD articles: is it really necessary?

P3, L50: 'surface mass balance (SMB)-elevation feedback' instead of 'surface-elevation feedback'. Please check this for the whole paper (e.g. title of section 2).

P3, L51: 'one dimension' instead of 'zero dimension'.

P3, L51: 'section 2' instead of 'section 1'.

P3, L53: 'section 3' instead of 'section 2'.

P3, L51: 'section 4' instead of 'section 3'.

P3, L55: 'feed' instead of 'enter'.

P5, L61: '(e.g.' instead of 'e.g. ('.

P5, L71: The authors already mention the Vialov profile above (L65), so there is no need to recall it.

P5, L73: Please define all quantities, i.e. $h$, $x$ and $L$ just after equation (1).

P5, L73: $h_m$ is more the surface elevation at the ice divide rather than the maximum surface elevation (Greve and Blatter, 2009).

P5, L74-75: Rephrase 'we do not aim for a realistic representation of the ice flow', which is not 'politically correct'.

P5, L77: 'constant and equal to surface accumulation' instead of 'homogeneous at a value, $a$'.

P5, L78: Define $L$ after equation (1) instead of here.

P5, L79: 'icebergs' instead of 'ice bergs'.

P5, L83-85: What is the purpose of writing down equations (2) and (3)? Mean surface elevation is not used at all in the study. If the authors demonstrate their usefulness, what is the derivative in equation (3)?

P6, L90: Precise which quantities you normalize.

P6, L93: 'equilibrium-line altitude (ELA)' instead of 'equilibrium line'.

P6, L108: We did not really understand how you 'rescaled' the SMB by $A$ in equation (7). Don't we miss $A$ in this equation, i.e. $h^m A - \gamma \Gamma h - a_0 = 0$?

P8, L121: Is it really necessary to have an entire section only for 13 lines? Wouldn't it be more useful to merge it with section 2?

P8, L122: 'ice sheet' instead of 'ice-sheet'.

P8, L124: Rephrase. Maybe: 'conditions, i.e. being a solution of the governing equation (7) and a minimum of the function...'

P8, L131: Equation (10) could be written more easily if starting by '$(1-m)$' instead of '$-(m-1)$'.

P9, L136-137: The first sentence is not really necessary since it was done in the previous section.

P9, first paragraph: Since you extensively compare your analysis to Ridley et al. (2010) and Robinson et al. (2012), maybe it would be useful to give us more insights about their methodology in the introduction (e.g. which models they use) and to try to provide an explanation for the differences between their models and the simple equation.

P9, L145: Why did you choose a threshold of 1.6°C? Is it only based on Ridley et al. (2010)?

P9, L145: What do you mean by 'both models'?

P9, L148: Define $\Delta T$: is it $T\text{-}T_c$ with $T$ being the temperature above the threshold?

P9, L158: You previously defined the melt rate ($\Delta a_0$) as negatively related to melting sensitivity and warming level (see L147)? And now it is positive. What is right?

P10, L170: 'with time, which' instead of 'with time which'.

P11, L188: 'choose' instead of 'chose'.

P11: The second and third paragraphs should be re-organized as they are a bit confusing: Figure 2 is explained only in the third paragraph but is already mentioned in the second paragraph.

P11, L200: 'translate' instead of 'translates'.

P11, L204: 'strongly depends' instead of 'depends strongly'.

P11, L204: 'threshold' instead of 'thresholds'.

P12, L211: Figure 4 is (almost) not discussed in the paper.

P12, L212-213: Rephrase, maybe: 'Since results obtained with equation (17) do not account for any dynamical discharge or even ice motion, they strongly deviate ...'.

P12, L215: It is not really apparent in Figure 3 that results deviate more strongly with a higher ice loss. Rephrase or rescale the figures.

P12, L221: Dynamic discharge is not as limited as suggested by different studies (see first specific comment). P13: Rewrite discussion taking into account all specific comments.

P13, L234: Rephrase 'For these curves in this figure'.

P13, L238: Precise that you mean sea-level contribution from the Greenland Ice Sheet.

P13, L241: 'dominant' instead of 'dominate'.

P13, L245: 'multi-millennial' instead of 'mult-millennial'.

P14, Tab. 1: Write down $\Delta T$ somewhere in the table.

P16, Fig. 2: 'median (...), likely (...) and very likely (...)' instead of 'median (...) and the likely (...) and very likely (...)'.

P17, Fig. 3: To be consistent with Fig. 2, it would be better to get the time in years (and not kyears).

P21, L338: Complete reference Greve (journal, year, volume, etc.).

P22, L347: Do we need to have the Howat TCD article since the TC article is listed in L342?

---

## Referee Comment (RC2) · Anonymous Referee #1 · 3 May 2016

We will resubmit a new version of our comments soon since some text is missing in the first two specific comments. We are sorry for the inconvenience.

---

## Referee Comment (RC3) · Anonymous Referee #1 · 3 May 2016

**General Comments**

This paper shows an interesting way of deriving the melt time of the Greenland Ice Sheet for different warming levels using a very simple approach based on three observable quantities: the equilibrium-line altitude (ELA), the atmospheric lapse rate and the melting sensitivity of the ice surface to temperature. The most interesting result is that the derived decay time quantitatively reproduces the range given by existing process-based numerical simulations. This study is relevant in the current context of Greenland Ice Sheet mass loss.

However, the approach suffers from several drawbacks that we detail in the Specific Comments below, especially the non-applicability of the decay time equation if dynamic discharge is taken into account, the lack of experiments to confirm the results given by the proposed equation, the lack of connection between different sections of the paper and the poor discussion. Therefore, we advise the authors to revise the paper either for providing a more substantial analysis of their work or for summarizing their results into a brief communication.

**Specific Comments**

1. The decay time equation proposed here does **not take into account ice dynamics**, as the authors state in section 5. However, a number of studies have shown that, even if the contribution of the dynamic part in Greenland ice loss seems to be less important than surface mass balance (SMB) changes, it is still quite substantial. One of the most recent modeling studies about this topic (Furst et al., 2015, TC) shows that 40% of the recent loss (2000-2010) is due to an increase in ice dynamic discharge (60% is due to SMB decrease). In terms of projections, using a 3D higher-order model with climate anomalies coming from 10 AOGCMs forced by the four RCPs climate scenarios, Furst et al. (2015) conclude that the sea-level rise of 1.4 to 16.6 cm by 2100 is predominantly caused by SMB decrease. They suggest the dynamic discharge contribution is limited by margin thinning and retreat as well as a competition between surface melting that removes ice before it reaches the calving front. Another modeling study based on four outlet glaciers that drain 22% of the Greenland Ice Sheet (Nick et al., 2013, Nature) shows that the dynamic contribution would be about 4-8.5 cm sea-level rise by 2100 versus 2.5-9.8 cm for SMB. Finally, radar (ERS-2) and laser (ICESat) altimetry observations show that mass changes in Greenland were dominated by SMB changes between 1995 and 2001, and then both SMB and dynamics equally contributed to the negative mass balance from 2001 to 2009 (Hurkmans et al., 2014, TC). Therefore, we think that not taking into account the dynamic part is a very strong assumption and we question the pertinence of the results presented here. At least, a scaling taking into account dynamics could be proposed in the decay time equation as well as a stronger discussion related to those three studies.

2. Even if we assume that ice loss only comes from SMB changes (which is the case of this study), the study **lacks some proofs that the decay time equation is robust against process-based studies**. Only Figure 2 clearly shows that the results agree well with two process-based numerical simulations, even if it does not show the time to lose 10% of ice for Ridley et al. (2010) under 1°C warming above threshold. Figure

3 shows the same quantity but for 50% of ice loss with only one numerical simulation (Robinson et al., 2012). What about Ridley et al. (2010) in Figure 3? In order to validate the simple equation proposed here, we think that the decay time for other $\alpha$ values (20%, 30%, 40%, 100%) should be shown along with results from process-based simulations.

3. It is not straightforward to understand **how sections 2 and 3 really fit** into the paper since the authors do not use the equations (1) to (10) related to the Vialov profile and the critical SMB for deriving the decay time equation (17), except equation (6) that relates surface melt rate and elevation. It is nice to see how the critical SMB and surface elevation below which a meltdown is inevitable are calculated but they are not really used in computing the main results of the paper (since the decay time only depends on the warming level, lapse rate, ELA and melting sensitivity). As far as we understood, one of the main purposes of sections 2 and 3 is to show where Figure 1 (which is quite nice) comes from.

4. The **discussion** clearly misses a robust analysis of the results. For example, some drawbacks related to the use of the decay time equation are presented at the end of section 5 but we think that they should really go into the discussion and be more detailed.

5. The whole paper talks about the surface-elevation feedback but in reality this is the **SMB-elevation feedback** (IPCC, 2013; Edwards et al., 2014, TC; Goelzer et al., 2013, J. Glaciology). Furthermore, the paper does not talk about the feedback of ice sheets in general but of the Greenland Ice Sheet in particular. Finally, the results of the paper focus less on the SMB-elevation feedback than on the melt time. Therefore, we suggest a different title: 'A simple equation for the melt time of the Greenland Ice Sheet'.

6. In **section 1 (Introduction)**, the first paragraph is very long and could be separated into two different paragraphs, one with the general Greenland ice loss context and the

other one with the temperature threshold and the SMB-elevation feedback. In any case, the link with the last paragraph of section 1 is not really done. We would add a clear explanation about the SMB-elevation feedback and the importance of determining the melt time for Greenland.

**Technical Corrections**

P2, L9: 'has been loosing' instead of 'is been loosing.

P2, L12: 'the' instead of 'The'.

P2, L13: Rephrase 'the lower the ice surface reaches into the atmosphere' since this is not clear.

P2, L14-15: The sentence 'The rate of ice loss is highly relevant for coastal protection worldwide' does not really fit here. It could go to the beginning or the end of the abstract.

P2, L16: Delete 'as it should be'.

P2, L16: Is the bit 'In order to contribute a little to the conceptual understanding' really needed? We would remove it.

P2, L18: We would cite the three observable 'characteristics', which we think are better defined as 'parameters'.

2, L20: 'critically depends' instead of 'depends critically'.

P2, L21: Use of 'critical' and 'critically' in the same sentence. Maybe replace 'critically' by 'strongly'.

P2, L21: 'the' instead of 'The'.

P2, L24: 'meltdown' instead of 'melt down'.

P3, L27: The first sentence is not totally accurate. Maybe: 'Global sea level rise has been raising in the past decades mainly due to ocean thermal expansion and melting ice (Church et al., 2013).' The last reference is more accurate than 'IPCC (2013)'.

P3, L28: 'past two decades' instead of 'two past decades'.

P3, L29-31: We think that some older references could be deleted and some newer

studies could be added, e.g. Kjeldsen et al. (2015, Nature) who study the Greenland ice loss since 1900 using aerial imagery, Khan et al. (2015, Reports on Progress in Physics) who provide a review of Greenland Ice Sheet mass balance, Shepherd et al. (2012) who provide results from the Ice sheet Mass Balance Inter-comparison Exercise (IMBIE).

P3, L36: 'Greve, 2000' instead of 'Greve, n.d.'.

P3, L38: 'critically depends' instead of 'depends critically'.

P3, L43: The authors need to agree whether they use 'meltdown' or 'melt-down' throughout the article (see also L24).

P3, L45: We did not find that Howat et al. (2014) mention a sea level rise contribution from Greenland of 7 m. Maybe Gregory et al. (2004, Nature Brief Communications) is a more suitable reference. Please also check your references for Howat et al. (2014) because you list both the TC and TCD articles: is it really necessary?

P3, L50: 'surface mass balance (SMB)-elevation feedback' instead of 'surface-elevation feedback'. Please check this for the whole paper (e.g. title of section 2).

P3, L51: 'one dimension' instead of 'zero dimension'.

P3, L51: 'section 2' instead of 'section 1'.

P3, L53: 'section 3' instead of 'section 2'.

P3, L51: 'section 4' instead of 'section 3'.

P3, L55: 'feed' instead of 'enter'.

P5, L61: '(e.g.' instead of 'e.g. ('.

P5, L71: The authors already mention the Vialov profile above (L65), so there is no need to recall it.

P5, L73: Please define all quantities, i.e. $h$, $x$ and $L$ just after equation (1).

P5, L73: $h_m$ is more the surface elevation at the ice divide rather than the maximum surface elevation (Greve and Blatter, 2009).

P5, L74-75: Rephrase 'we do not aim for a realistic representation of the ice flow', which is not 'politically correct'.

P5, L77: 'constant and equal to surface accumulation' instead of 'homogeneous at a value, $a$'.

P5, L78: Define L after equation (1) instead of here.

P5, L79: 'icebergs' instead of 'ice bergs'.

P5, L83-85: What is the purpose of writing down equations (2) and (3)? Mean surface elevation is not used at all in the study. If the authors demonstrate their usefulness, what is the derivative in equation (3)?

P6, L90: Precise which quantities you normalize.

P6, L93: 'equilibrium-line altitude (ELA)' instead of 'equilibrium line'.

P6, L108: We did not really understand how you 'rescaled' the SMB by $A$ in equation (7). Don't we miss $A$ in this equation, i.e. $h^m A - \gamma\Gamma h - a_0 = 0$?

P8, L121: Is it really necessary to have an entire section only for 13 lines? Wouldn't it be more useful to merge it with section 2?

P8, L122: 'ice sheet' instead of 'ice-sheet'.

P8, L124: Rephrase. Maybe: 'conditions, i.e. being a solution of the governing equation (7) and a minimum of the function...'

P8, L131: Equation (10) could be written more easily if starting by '$(1-m)$' instead of '$-(m-1)$'.

P9, L136-137: The first sentence is not really necessary since it was done in the previous section.

P9, first paragraph: Since you extensively compare your analysis to Ridley et al. (2010) and Robinson et al. (2012), maybe it would be useful to give us more insights about their methodology in the introduction (e.g. which models they use) and to try to provide an explanation for the differences between their models and the simple equation.

P9, L145: Why did you choose a threshold of 1.6°C? Is it only based on Ridley et al. (2010)?

P9, L145: What do you mean by 'both models'?

P9, L148: Define $\Delta T$: is it $T$-$T_c$ with $T$ being the temperature above the threshold?

P9, L158: You previously defined the melt rate ($\Delta a_0$) as negatively related to melting sensitivity and warming level (see L147)? And now it is positive. What is right?

P10, L170: 'with time, which' instead of 'with time which'.

P11, L188: 'choose' instead of 'chose'.

P11: The second and third paragraphs should be re-organized as they are a bit confusing: Figure 2 is explained only in the third paragraph but is already mentioned in the second paragraph.

P11, L200: 'translate' instead of 'translates'.

P11, L204: 'strongly depends' instead of 'depends strongly'.

P11, L204: 'threshold' instead of 'thresholds'.

P12, L211: Figure 4 is (almost) not discussed in the paper.

P12, L212-213: Rephrase, maybe: 'Since results obtained with equation (17) do not account for any dynamical discharge or even ice motion, they strongly deviate ...'.

P12, L215: It is not really apparent in Figure 3 that results deviate more strongly with a higher ice loss. Rephrase or rescale the figures.

P12, L221: Dynamic discharge is not as limited as suggested by different studies (see first specific comment).

P13: Rewrite discussion taking into account all specific comments.

P13, L234: Rephrase 'For these curves in this figure'.

P13, L238: Precise that you mean sea-level contribution from the Greenland Ice Sheet.

P13, L241: 'dominant' instead of 'dominate'.

P13, L245: 'multi-millennial' instead of 'mult-millennial'.

P14, Tab. 1: Write down $\Delta T$ somewhere in the table.

P16, Fig. 2: 'median (...), likely (...) and very likely (...)' instead of 'median (...) and the likely (...) and very likely (...)'.

P17, Fig. 3: To be consistent with Fig. 2, it would be better to get the time in years (and not kyears).

P21, L338: Complete reference Greve (journal, year, volume, etc.).

P22, L347: Do we need to have the Howat TCD article since the TC article is listed in L342?

---

## Short Comment (SC2) · 4 May 2016

Dear Dr Fettweis,

We would like to thank the reviewer for taking the time to work so intensely with our manuscript and we are very happy that the reviewer considers our work interesting. We agree with the reviewer that the comparison of the simple equation that we derive with the process-based models is interesting, but we do not really consider it the "most interesting result" of the paper, which is why we have not made it the title of the study.

As much as we appreciate the reviewer's comments, we would also very much appreciate if we could keep the paper in the spirit in which we chose it to be. We fully recognize that there are different views on what is important and relevant science. The reviewer has one take on this and we have a slightly different one. A number of the specific

comments of the reviewer point into the direction of changing the spirit of the paper. We would very much appreciate if the editor would allow us to keep the essence of the paper as we intended it to be: An equation that captures only one specific aspect of the melting of an ice sheet. It is purposefully simple. It purposefully does not include any dynamic effects and it is purposefully not a modelling study. It is a paper on a simple piece of theory that we did not find in the literature and would like to report here so that it can be used by others in the future.

There is no doubt that our study is not a comprehensive analysis of ice sheet mass loss and we make this very clear throughout the paper. We believe that there is a special merit in extracting specific processes from complex physical phenomena and this is one small contribution in this direction. We hope that it will be of interest to other researchers in the field and believe that it can be used without further complication.

Having said that we very much appreciate the reviewer's suggestions on the manuscript and we will, of course, incorporate and address every aspect that the reviewer raises - in particular, where it helps to improve the manuscript's clarity and legibility. If that means that it becomes a "brief communication", that would be fine. We just thought that there is no need to reduce the number of figures and references in order to meet the length limitations of a brief communication, but if that is required, we can do this, of course.

We would again like to thank the reviewer for the detailed review of our manuscript and will give a detailed response to all of the reviewer's comments together with a revised manuscript that also include the second reviewer's comments.

Best wishes, Anders Levermann

---

## Referee Comment (RC4) · Anonymous Referee #2 · 7 May 2016

The authors derive a simple relationship for the elevation-melt feedback based on an analytical 1D flowline profile model of an ice sheet. The relationship allows calculation of the critical mean height that leads to melting of the ice sheet, given a prescribed rate of accumulation that changes with height and temperature anomaly. It also allows the decay time to be estimated, which compares well with more complex models for the Greenland ice sheet domain – serving as a validation of the approach. The method is described well and looks promising for distilling this feedback into a simple relationship. I would therefore only suggest some minor changes before publication.

The decay time equation is interesting and could certainly be useful for risk management planning. However, it seems that its validity is questionable for higher volume losses, ie, 50% or more. The authors discuss this briefly, but then the tables give decay times for both 50% and 100% volume loss as if it has equal weight to the 10% loss

time estimates. I would recommend differentiating these results somehow, and emphasizing that this approach is more useful for diagnosing the earlier stages of decline.

In addition, I am missing the transition from height reduction to volume loss. The fraction $\alpha$ is introduced to represent the volume loss, but as I understand it, this is applied interchangeably with the mean height reduction. A justification of why mean(H) = V is needed.

Finally, I realize the authors are interested in promoting this as a tool for risk assessment, but I think the manuscript would benefit more from discussion of the theoretical implications of the approach. What does the form of your equation mean in terms of the process(es) represented? Why does the rate of fastest melting saturate (Fig. 4) – what in the equations limits the time scale of melt? When could this be violated?

Minor comments

Title: As with the first reviewer, "Surface-elevation" feedback seems incorrect. Either "Temperature-elevation" or more likely "SMB-elevation" feedback makes more sense to me.

Page 2, line 9: is been losing => has been losing

Page 2, line 16: Suggest deleting "as it should be" and "a little".

Page 3, line 46: The sentence starting with "The framework" seems to belong to a new paragraph. Furthermore, so far, no framework has been introduced so it seems out of place without a bit of introduction.

Page 3, line 47: imclude => include

Page 3, line 51: dimension => dimensions

Page 6, line 98: Consider rephrasing "observed". This is open to debate.

Page 7, line 112: melt-down => melt. Also note that the lower branch (ie, the melted

state) is also a stable state, as shown in Fig. 1. Consider rephrasing slightly.

Page 8, line 132: It seems that this point "a_0 to decline linearly" should be mentioned earlier. This is in fact a pretty critical assumption to the whole approach, no?

Page 8, line 133: off-set => offset

Page 12, line 225: can thus be used when => can thus be used if

Page 13, line 241: dominate => dominant

Page 13, line 243: the 015 Paris => the 2015 Paris

Page 15, line 254: melt-down => melt

Figure 4: Grid lines would help to be able to compare the panels. It would also be easier if they were presented in a vertical column, to emphasize the shift in time scale for higher temperatures.
* * *

---

## Short Comment (SC4) · 8 May 2016

Dear Editor,

we very much appreciate the reviewer's comments which are very constructive and helpful. We will incorporate them all.

We actually did not mean to use this approach for risk assessment and will make sure that this misunderstanding is eliminated in the revised manuscript.

Best wishes, Anders Levermann

---

## Author Response (AR1)

Dear Dr. Fettweis,

Thank you for handling the review process of our manuscript. We are very happy that both reviewers appreciate the novelty of our study and consider it interesting. We present here a detailed response to each of the reviewers' comments *in italic and blue*.

We highly appreciate that both reviewers have taken so much time and effort in reviewing our study. The manuscript has improved significantly thanks to their corrections and suggestions. However, we must admit that we are slightly frustrated by some of the very confident, strong and strict demands made by reviewer #1, because they seem to be based in places on a misunderstanding of the theory (e.g. concerning equations (3) and (4)) and in other places simply on a different appreciation for theory and a different understanding of the purpose of this study than we have. In particular, we could not meet the demand that we make the computation of the melt time the central aspect of our study. There exist complex numerical models which are much better suited to address this question and our approach is not comprehensive in the number and detail of the processes that are represented – on the contrary, it is purposefully simple in order to emphasize what can be explained by the melt-elevation feedback and what physical characteristics this feedback has.

That being said, we really appreciate that both reviewers have provided such a thorough review of the manuscript, including the correction of typos and other issues, and we took on very seriously that he or she finds that the discussion was insufficient and have sought to improve this in the revised manuscript - detailed answers to all points raised by the reviewer are given below.

We are very sorry that we were not able to comply with all of reviewer #1's requests. Generally we try to do this, but it was not possible in this case. We hope that you and the reviewers will never-the-less agree with us that while our paper does not solve all matters of ice sheet melting, it is still useful and interesting for the readers of The Cryosphere.

Best wishes,

Anders and Ricarda

**Dear Dr. Fettweis,**

**Thank you for handling the review process of our manuscript. We are very happy that both reviewers appreciate the novelty of our study and consider it interesting. We think that the manuscript has improved significantly due to their corrections and suggestions. We present here a detailed response to each of the reviewers' comments** *in italic and blue*.

**Reviewer #1:**

**General Comments**

This paper shows an interesting way of deriving the melt time of the Greenland Ice Sheet for different warming levels using a very simple approach based on three observable quantities: the equilibrium-line altitude (ELA), the atmospheric lapse rate and the melting sensitivity of the ice surface to temperature. The most interesting result is that the derived decay time quantitatively reproduces the range given by existing process-based numerical simulations. This study is relevant in the current context of Greenland Ice Sheet mass loss. However, the approach suffers from several drawbacks that we detail in the Specific Comments below, especially the non-applicability of the decay time equation if dynamic discharge is taken into account, the lack of experiments to confirm the results given by the proposed equation, the lack of connection between different sections of the paper and the poor discussion. Therefore, we advise the authors to revise the paper either for providing a more substantial analysis of their work or for summarizing their results into a brief communication.

*Response:*

*We would like to thank the reviewer for taking the time to work so intensely with our manuscript and we are very happy that the reviewer considers our work interesting. We agree with the reviewer that the comparison of the simple equation that we derive with the complex process-based models is interesting, but we do not consider this the "most interesting result" of the paper.As reflected in our choice of the title we seek to provide the simplest possible representation of the melt-elevation feedback and derive some (hopefully) curious characteristics of the resulting mini-theory.*

*Some of the specific comments of the reviewer (below) aim at changing this spirit of the study. We fully appreciate that there are different views on what is important and relevant science. The reviewer has one take on this and we have our own. As much as we appreciate the reviewer's comments, we would also very much appreciate if we could keep the general nature of our paper as it was intended: An equation that captures one specific and important aspect of the large-scale melting of an ice sheet.*

*Our study is purposefully simple. It purposefully does not include any ice dynamic effects and it is purposefully not a modelling study. It is a simple theory paper. There is no doubt that our study is not a comprehensive analysis of ice-sheet mass-loss and we make this very clear throughout the paper. We believe that there is special merit in extracting specific processes from complex physical phenomena and this is one small contribution in this direction. We hope*

*and believe that it will be of interest to other researchers in the field and that it can be used as a conceptual approach to further understanding the melt-elevation feedback.*

*Having said that we very much appreciate the reviewer's suggestions on the manuscript and we will, of course, address every aspect that the reviewer raises. We think that the manuscript's clarity and legibility have improved significantly thanks to the many helpful suggestions.*

*We will give a detailed response to all of the reviewer's comments together with a revised manuscript that also includes the second reviewer's comments.*

**Specific Comments**

1. The decay time equation proposed here does **not take into account ice dynamics**, as the authors state in section 5. However, a number of studies have shown that, even if the contribution of the dynamic part in Greenland ice loss seems to be less important than surface mass balance (SMB) changes, it is still quite substantial. One of the most recent modeling studies about this topic (Furst et al., 2015, TC) shows that 40% of the recent loss (2000-2010) is due to an increase in ice dynamic discharge (60% is due to SMB decrease). In terms of projections, using a 3D higher-order model with climate anomalies coming from 10 AOGCMs forced by the four RCPs climate scenarios, Furst et al. (2015) conclude that the sea-level rise of 1.4 to 16.6 cm by 2100 is predominantly caused by SMB decrease. They suggest the dynamic discharge contribution is limited by margin thinning and retreat as well as a competition between surface melting that removes ice before it reaches the calving front. Another modeling study based on four outlet glaciers that drain 22% of the Greenland Ice Sheet (Nick et al., 2013, Nature) shows that the dynamic contribution would be about 4-8.5 cm sea-level rise by 2100 versus 2.5-9.8 cm for SMB. Finally, radar (ERS-2) and laser (ICESat) altimetry observations show that mass changes in Greenland were dominated by SMB changes between 1995 and 2001, and then both SMB and dynamics equally contributed to the negative mass balance from 2001 to 2009 (Hurkmans et al., 2014, TC). Therefore, we think that not taking into account the dynamic part is a very strong assumption and we question the pertinence of the results presented here. At least, a scaling taking into account dynamics could be proposed in the decay time equation as well as a stronger discussion related to those three studies.

*Response:*

*There is no doubt that ice dynamics is important for both ice sheets on Antarctica and Greenland. However, as explained above, this paper is not about being comprehensive. For example, the very important articles that derived the Shallow Ice Approximation made the very strong assumption of zero ice velocity at the ground which is not the case in most regions in Antarctica that are crucial for the ice loss of the continent. The Shallow Ice Approximation was nevertheless a very important contribution to glaciological theory. While we cannot claim that our approach is even remotely as relevant as that, we are convinced that extracting only one specific feedback from a complex problem is helpful for the understanding, and may it only be theoretical. It is for example curious to view the melt-elevation feedback (as we now call it in*

*response to both reviewers' requests) as a linear response function with an increasing, not decreasing long-term tail. It is also curious to see the "critical slowing down" that can be observed in many non-linear systems near their threshold. These two phenomena would, for example, be diluted and become less clear by adding more processes to the equation.*

*We understand that the reviewer thinks that our discussion of previous work was not sufficient (although we cite more than 60 articles while in a brief communication only 20 references would be allowed). In order to give a more comprehensive discussion we have added additional references (including the ones mentioned by the reviewer) and their discussion.*

2. Even if we assume that ice loss only comes from SMB changes (which is the case of this study), the study **lacks some proofs that the decay time equation is robust against process-based studies**. Only Figure 2 clearly shows that the results agree well with two process-based numerical simulations, even if it does not show the time to lose 10% of ice for Ridley et al. (2010) under 1_C warming above threshold. Figure3 shows the same quantity but for 50% of ice loss with only one numerical simulation (Robinson et al., 2012). What about Ridley et al. (2010) in Figure 3? In order to validate the simple equation proposed here, we think that the decay time for other values (20%, 30%, 40%, 100%) should be shown along with results from process-based simulations.

**Response:**

*This comment reflects the disagreement that we have with the reviewer with respect to the purpose of our study. We would like to emphasize that this is __not__ a modelling study, but a simple piece of theory. We were searching for an analysis of the melt-elevation feedback in the same spirit as they have been carried out by a number of authors in other contexts (e.g. Gnanadesikan, Science, 1999; Stommel, 1961, Levermann et al. PNAS 2009 and a number of publications by J. Oerlemans in Nature and Science etc.). Since we did not find this kind of equation we derived it, analyzed some interesting mathematical properties of the solution (see above) and compared it to some available model results. In fact, there are not too many model simulations that increase the temperature on Greenland by a constant value and report the result for long enough to lose 10% of the ice sheet (we only know of Ridley et al. 2010 and Robinson et al. 2012).*

*The Figure 3 of Ridley et al. (2010) that the reviewer refers to, can be compared with our results in the sense that it obviously shows that there is more complex physics at play in the numerical model than there is in our simple model. That is the reason why Ridley et al. find a multi-stability of the ice sheet while we only find a bi-stability. Ridley et al. show that this multi-stability arises predominantly from horizontal differences in topography and surface-mass balance which is why some parts of the ice sheet are more and some are less sensitive to a surface temperature increase. This is not surprising and it is not at odds with our equation. It is just not captured by a simple conceptual model without horizontal resolution.*

3. It is not straightforward to understand **how sections 2 and 3 really fit** into the paper since the authors do not use the equations (1) to (10) related to the Vialov profile and the critical SMB for deriving the decay time equation (17), except equation (6) that relates surface melt rate and elevation. It is nice to see how the critical SMB and surface elevation below which a meltdown is inevitable are calculated but they are not really used in computing the main results of the paper (since the decay time only depends on the warming level, lapse rate, ELA and melting sensitivity). As far as we understood, one of the main purposes of sections 2 and 3 is to show where Figure 1 (which is quite nice) comes from.

*Response:*

*We believe that this issue arises because the reviewer views the purpose of the paper as providing a means to estimate the decay time of the Greenland Ice Sheet. While this is an interesting application, it is not the purpose of the manuscript. Instead we aim to provide a simple conceptual theory for the decay of an ice sheet due to the melt-elevation feedback beyond the critical temperature threshold. We believe that to this end it is interesting for the reader to see a derivation of this threshold and a computation of the threshold position as a starting point for the temporal equation. We have added sentences to explain the purpose of these sections to the introduction and each of the sections to clarify.*

4. The **discussion** clearly misses a robust analysis of the results. For example, some drawbacks related to the use of the decay time equation are presented at the end of section 5 but we think that they should really go into the discussion and be more detailed.

*Response:*

*We have shifted parts of the discussion in section 5 to the discussion section and expanded it. We hope that it is now clearer that we are presenting a conceptual model and not a method to comprehensively estimate the melt time of the Greenland Ice Sheet.*

5. The whole paper talks about the surface-elevation feedback but in reality this is the **SMB-elevation feedback** (IPCC, 2013; Edwards et al., 2014, TC; Goelzer et al., 2013, J. Glaciology). Furthermore, the paper does not talk about the feedback of ice sheets in general but of the Greenland Ice Sheet in particular. Finally, the results of the paper focus less on the SMB-elevation feedback than on the melt time. Therefore, we suggest a different title: 'A simple equation for the melt time of the Greenland Ice Sheet'.

*Response:*

*We changed the name of the feedback to "melt-elevation feedback" in response to both reviewers' comments. As we explained above, we disagree with the reviewer on the purpose of the study. Since we consider it to be a conceptual paper that is relevant for all ice sheets in general we would like to keep the title with the change in the feedback's name. We think this is*

*reflected in the fact that all but one of the six sections of the paper are general in nature and only one section deals specifically with Greenland as an example.*

6. In **section 1 (Introduction)**, the first paragraph is very long and could be separated into two different paragraphs, one with the general Greenland ice loss context and the other one with the temperature threshold and the SMB-elevation feedback. In any case, the link with the last paragraph of section 1 is not really done. We would add a clear explanation about the SMB-elevation feedback and the importance of determining the melt time for Greenland.

*Response:*

*Done.*

**Technical Corrections**

P2, L9: 'has been loosing' instead of 'is been loosing.

*Response: Done.*

P2, L12: 'the' instead of 'The'.

*Response: Done.*

P2, L13: Rephrase 'the lower the ice surface reaches into the atmosphere' since this is not clear.

*Response: Changed to: "the more ice is lost, the lower the ice surface and the warmer the surface air temperature which fosters further melting and ice loss."*

P2, L14-15: The sentence 'The rate of ice loss is highly relevant for coastal protection worldwide' does not really fit here. It could go to the beginning or the end of the abstract.

*Response: The sentence was moved to the beginning of the abstract.*

P2, L16: Delete 'as it should be'.

*Response: Done*

P2, L16: Is the bit 'In order to contribute a little to the conceptual understanding' really needed?  We would remove it.

*Response: We have rephrased this part to make our intensions clearer:*

*"The computation of this rate so far relies on process-based numerical models which are the appropriate tools to capture the complexity of the problem. By contrast, we aim here at gaining conceptual understanding by deriving a purposefully simple equation for the self-enforcing feedback and use it to estimate the melt time for different levels of warming using three observable characteristics of the ice sheet itself and its surroundings."*

P2, L18: We would cite the three observable 'characteristics', which we think are better defined as 'parameters'.

*Response: We have defined the characteristics of the ice sheet that we use in the main text. We cannot see why the word "parameters" is better defined than "characteristics".*

P2, L20: 'critically depends' instead of 'depends critically'.

*Response: We have checked with a native speaker that holds a master degree in English literature and were assured that this is not a grammatical error, but a matter of choice. In accordance with our advisor's opinion, we prefer our original version of the wording.*

P2, L21: Use of 'critical' and 'critically' in the same sentence. Maybe replace 'critically' by 'strongly'.

*Response: Done.*

P2, L21: 'the' instead of 'The'.

*Response: Done.*

P2, L24: 'meltdown' instead of 'melt down'.

*Response: Done.*

P3, L27: The first sentence is not totally accurate. Maybe: 'Global sea level rise has been raising in the past decades mainly due to ocean thermal expansion and melting ice (Church et al., 2013).' The last reference is more accurate than 'IPCC (2013)'.

*Response: Yes, thank you - we changed the sentence to "In past decades global mean sea level has been rising mainly by expansion of ocean waters and melting of ice on land (Church et al. 2013)."*

P3, L28: 'past two decades' instead of 'two past decades'.

*Response: Done.*

P3, L29-31: We think that some older references could be deleted and some newer studies could be added, e.g. Kjeldsen et al. (2015, Nature) who study the Greenland ice loss since 1900 using aerial imagery, Khan et al. (2015, Reports on Progress in Physics) who provide a review of Greenland Ice Sheet mass balance, Shepherd et al. (2012) who provide results from the Ice sheet Mass Balance Inter-comparison Exercise (IMBIE).

*Response: We appreciate the reviewer's suggestions and have happily added them to our reference list. We prefer not to delete any older references since these earlier studies (between 2011 and 2016) are still highly relevant as is the Shepherd et al. 2012 study suggested by the reviewer.*

P3, L36: 'Greve, 2000' instead of 'Greve, n.d.'.

*Response: Done.*

P3, L38: 'critically depends' instead of 'depends critically'.

*Response: See our comment above (P2, L20).*

P3, L43: The authors need to agree whether they use 'meltdown' or 'melt-down' throughout the article (see also L24).

*Response: Done.*

P3, L45: We did not find that Howat et al. (2014) mention a sea level rise contribution from Greenland of 7 m. Maybe Gregory et al. (2004, Nature Brief Communications) is a more suitable reference. Please also check your references for Howat et al. (2014) because you list both the TC and TCD articles: is it really necessary?

*Response: No, this was a mistake. Thanks for the hint.*

P3, L50: 'surface mass balance (SMB)-elevation feedback' instead of 'surfaceelevation feedback'. Please check this for the whole paper (e.g. title of section 2).

*Response: Since "surface mass balance-elevation feedback" seems very long, we would like to follow the second reviewer and denote it as "melt-elevation feedback". We hope this is alright with the editor and both reviewers. We have changed this through-out the manuscript.*

P3, L51: 'one dimension' instead of 'zero dimension'.

*Response: Done.*

P3, L51: 'section 2' instead of 'section 1'.

*Response: Done.*

P3, L53: 'section 3' instead of 'section 2'.

*Response: Done.*

P3, L51: 'section 4' instead of 'section 3'.

*Response: Done.*

P3, L55: 'feed' instead of 'enter'.

*Response: We would like to keep "enter" because the verb "feed" would suggest that the model is really a means to transform parameters into melt rates and that is (as we stated) not the purpose of the model.*

P5, L61: '(e.g.' instead of 'e.g. ('.

*Response: Done.*

P5, L71: The authors already mention the Vialov profile above (L65), so there is no need to recall it.

*Response: We would like to keep it here. It is just a half-sentence and it allows the reader to skip the introduction and still understand the following chapters.*

P5, L73: Please define all quantities, i.e. h, x and L just after equation (1).

*Response: Done.*

P5, L73: $h_m$ is more the surface elevation at the ice divide rather than the maximum surface elevation (Greve and Blatter, 2009).

*Response: As can be seen from the equation it is also the maximum elevation of the ice sheet.*

P5, L74-75: Rephrase 'we do not aim for a realistic representation of the ice flow', which is not 'politically correct'.

*Response: Done.*

P5, L77: 'constant and equal to surface accumulation' instead of 'homogeneous at a value, a'.

*Response: Done.*

P5, L78: Define L after equation (1) instead of here.

*Response: Done.*

P5, L79: 'icebergs' instead of 'ice bergs'.

*Response: Done.*

P5, L83-85: What is the purpose of writing down equations (2) and (3)? Mean surface elevation is not used at all in the study. If the authors demonstrate their usefulness, what is the derivative in equation (3)?

*Response: We consider it interesting to the reader to relate the maximum surface elevation to the mean surface elevation, both of which characterize the ice-sheet geometry in the Vialov profile. Since this is not a long derivation we would like to keep it. There is no derivative in equation (3), so we are confused as to what the reviewer refers to?*

P6, L90: Precise which quantities you normalize.

*Response: They are given in the same sentence and refer to the surface elevation, surface mass balance and the ice softness.*

P6, L93: 'equilibrium-line altitude (ELA)' instead of 'equilibrium line'.

*Response: Done.*

P6, L108: We did not really understand how you 'rescaled' the SMB by A in equation (7). Don't we miss A in this equation, i.e. $hmA - \nabla h - a0 = 0$?

*Response: We have added a sentence with an additional equation to explain the rescaling.*

P8, L121: Is it really necessary to have an entire section only for 13 lines? Wouldn't it be more useful to merge it with section 2?

*Response: We merged it with section 2.*

P8, L122: 'ice sheet' instead of 'ice-sheet'.

*Response: Done.*

P8, L124: Rephrase. Maybe: 'conditions, i.e. being a solution of the governing equation (7) and a minimum of the function...'

*Response: Done.*

P8, L131: Equation (10) could be written more easily if starting by '(1-m)' instead of '-(m-1)'.

*Response: Thanks for the advice. Since m is generally larger than 1 we prefer this form so that it can be seen immediately that $a_{0c}$ and that the exponent is positive, i.e. the $a_{0c}$ is an increasing and not decreasing function of Γ and γ.*

P9, L136-137: The first sentence is not really necessary since it was done in the previous section.

*Response: We reformulated in order to meet this comment by the reviewer as well as the earlier request to explain the role of the former sections 2 and 3 (which are now merged into section 2).*

P9, first paragraph: Since you extensively compare your analysis to Ridley et al. (2010) and Robinson et al. (2012), maybe it would be useful to give us more insights about their methodology in the introduction (e.g. which models they use) and to try to provide an explanation for the differences between their models and the simple equation.

*Response: Done.*

P9, L145: Why did you choose a threshold of 1.6_C? Is it only based on Ridley et al. (2010)?

*Response: Robinson et al (2012) provide a number for their threshold which is 1.6°C for the parameter setting that we used. Ridley et al. (2010) carried out fewer simulations and a value of 1.6°C is consistent with their results, but no precise threshold value can be derived from their publication. Since this part of the paper is merely an illustration of the possible application of the theory, we consider it best to use the Robinson value of 1.6°C and state this clearly and transparently. We have added a half-sentence to clarify.*

P9, L145: What do you mean by 'both models'?

*Response: The Robinson et al. and the Ridley et al model. We hope this has become clearer by the addition of the half-sentence.*

P9, L148: Define _T: is it T-Tc with T being the temperature above the threshold?

*Response: Yes. We have added the definition.*

P9, L158: You previously defined the melt rate (_a0) as negatively related to melting sensitivity and warming level (see L147)? And now it is positive. What is right?

*Response: Thank you very much for spotting this typo again. The negative value is correct and it has been corrected here.*

P10, L170: 'with time, which' instead of 'with time which'.

*Response: Done.*

P11, L188: 'choose' instead of 'chose'.

*Response: Done.*

P11: The second and third paragraphs should be re-organized as they are a bit confusing: Figure 2 is explained only in the third paragraph but is already mentioned in the second paragraph.

*Response: We would like to thank the reviewer for this advice. We have shifted the second paragraph down in the section in order to keep the flow of the text, i.e. first explain what is done*

*in order to obtain the simple model results including their uncertainty range and then add the complex model results. We hope the text is now clearer.*

P11, L200: 'translate' instead of 'translates'.

*Response: Done.*

P11, L204: 'strongly depends' instead of 'depends strongly'.

*Response: Done.*

P11, L204: 'threshold' instead of 'thresholds'.

*Response: Done.*

P12, L211: Figure 4 is (almost) not discussed in the paper.

*Response: We have expanded the discussion of Figure 4.*

P12, L212-213: Rephrase, maybe: 'Since results obtained with equation (17) do not account for any dynamical discharge or even ice motion, they strongly deviate ...'.

*Response: We would like to start the paragraph with a clear statement introducing the discussion. To this end we find the sentence "The simple equation provided here is clearly limited in its applicability." very clear and would like to keep it. The content is the same as suggested by the reviewer.*

P12, L215: It is not really apparent in Figure 3 that results deviate more strongly with a higher ice loss. Rephrase or rescale the figures.

*Response: We agree with the reviewer that this formulation was misleading. We rephrased the sentence to say that it can be clearly seen that the **functional form** is not captured by our simple equation. We believe that this is indeed the case.*

P12, L221: Dynamic discharge is not as limited as suggested by different studies (see first specific comment).

*Response: We have reformulated to say "Some studies suggest".*

P13: Rewrite discussion taking into account all specific comments.

*Response: We have changed the discussion in accordance with the changes made.*

P13, L234: Rephrase 'For these curves in this figure'.

*Response: Done.*

P13, L238: Precise that you mean sea-level contribution from the Greenland Ice Sheet.

*Response: Done.*

P13, L241: 'dominant' instead of 'dominate'.

*Response: Done.*

P13, L245: 'multi-millennial' instead of 'mult-millennial'.

*Response: Done.*

P14, Tab. 1: Write down _T somewhere in the table.

*Response: We do not understand the request.*

P16, Fig. 2: 'median (...), likely (...) and very likely (...)' instead of 'median (...) and the likely (...) and very likely (...)'.

*Response: Done.*

P17, Fig. 3: To be consistent with Fig. 2, it would be better to get the time in years (and not kyears).

*Response: We have changed the numbers in Fig. 3 to kyrs in order to be consistent.*

P21, L338: Complete reference Greve (journal, year, volume, etc.).

*Response: Done.*

P22, L347: Do we need to have the Howat TCD article since the TC article is listed in L342?

*Response: No, this was a mistake. Thanks for the hint.*

**Reviewer #2:**

The authors derive a simple relationship for the elevation-melt feedback based on an analytical 1D flowline profile model of an ice sheet. The relationship allows calculation of the critical mean height that leads to melting of the ice sheet, given a prescribed rate of accumulation that changes with height and temperature anomaly. It also allows the decay time to be estimated, which compares well with more complex models for the Greenland ice sheet domain – serving as a validation of the approach. The method is described well and looks promising for distilling this feedback into a simple relationship.

**I would therefore only suggest some minor changes before publication**. The decay time equation is interesting and could certainly be useful for risk management planning. However, it seems that its validity is questionable for higher volume losses, ie, 50% or more. The authors discuss this briefly, but then the tables give decay times for both 50% and 100% volume loss as if it has equal weight to the 10% loss time estimates. I would recommend differentiating these results somehow, and emphasizing that this approach is more useful for diagnosing the earlier stages of decline.

In addition, I am missing the transition from height reduction to volume loss. The fraction _ is introduced to represent the volume loss, but as I understand it, this is applied interchangeably with the mean height reduction. A justification of why mean(H) = V is needed.

Finally, I realize the authors are interested in promoting this as a tool for risk assessment, but I think the manuscript would benefit more from discussion of the theoretical implications of the approach. What does the form of your equation mean in terms of the process(es) represented? Why does the rate of fastest melting saturate (Fig. 4) – what in the equations limits the time scale of melt? When could this be violated?

*Response: We very much appreciate the reviewer's positive assessment and are grateful for the constructive comments. We are sorry that we missed to provide the translation from height to volume. In fact we just assumed a constant area by which the percentage change in ice thickness translates directly to the percentage change in ice volume. We have now added this to the paper in section 3 (end of first paragraph) together with another statement that the quantitative interpretation of the results is subject to a number of limitations and that the results are supposed to be taken conceptual in nature.*

*In fact we do not plan to use this study for any risk analysis and do not think that it should be used to this end. We have added a sentence concerning this issue to the abstract and are sorry if we gave the impression that the method can be used for risk assessments. Given the clear limitations, including the exclusion of dynamic effects and the strong assumption that the ice height directly translates into ice volume through a constant area, forbid this kind of quantitative interpretation. We hope this has now become clearer.*

*With respect to the distinction of 10%, 50% and 100% volume loss, we do not fully know how to handle this issue. At the moment we say that 10% is the most useful regime in which the method gives relatively good representations of the complex models. This has two reasons: for one it is enough volume change so that the melt-elevation feedback is indeed significant and secondly the time that elapsed during the melt was not too long for ice dynamic effects (at least in the complex models that we used for comparison) to become dominant. We say this in the*

*text. We would like to keep the table with all percentages because we believe that it is interesting to see how long a complete melting without ice dynamics and without horizontal distinction would take. But this is not crucial. If the reviewer or editor prefer to take out the part of the table, we will definitely do this. For the time being we have added a sentence to the caption of the table stating that the 10% values have some quantitative merit while the other numbers cannot be taken as valid estimates of real world ice loss time scales.*

**Minor comments**

Title: As with the first reviewer, "Surface-elevation" feedback seems incorrect. Either "Temperature-elevation" or more likely "SMB-elevation" feedback makes more sense to me.

*Response: We are happy to change the name of the feedback. In order not to make it too long, we would like to use the term melt- elevation feedback, if that is alright.*

Page 2, line 9: is been losing => has been losing

*Response: Done.*

Page 2, line 16: Suggest deleting "as it should be" and "a little".

*Response: Done.*

Page 3, line 46: The sentence starting with "The framework" seems to belong to a new paragraph. Furthermore, so far, no framework has been introduced so it seems out of place without a bit of introduction.

*Response: The sentence was shifted to the end of the section so that the framework is mentioned prior to this statement.*

Page 3, line 47: imclude => include

*Response: Done.*

Page 3, line 51: dimension => dimensions

*Response: Done.*

Page 6, line 98: Consider rephrasing "observed". This is open to debate.

*Response: We rephrased to "commonly chosen to be".*

Page 7, line 112: melt-down => melt. Also note that the lower branch (ie, the melted state) is also a stable state, as shown in Fig. 1. Consider rephrasing slightly.

*Response: We agree with the reviewer that physically there is a third solution with H=0. Since this solution is not capture mathematically by the theory we believe that the current formulation is correct, isn't it?*

Page 8, line 132: It seems that this point "a_0 to decline linearly" should be mentioned earlier. This is in fact a pretty critical assumption to the whole approach, no?

*Response: We fully agree and have mentioned this now on page 7 where Fig. 1 is first mentioned. In fact we could just change the x-axis of the figures to be a_0, but we thought it*

*would be more illustrative to show the dependence on temperature change. In this sense there is no "loss of generality", but we agree that it has to be made very clear and we hope that this has now become clearer.*

Page 8, line 133: off-set => offset

*Response: Done.*

Page 12, line 225: can thus be used when => can thus be used if

*Response: Done.*

Page 13, line 241: dominate => dominant

*Response: Done.*

Page 13, line 243: the 015 Paris => the 2015 Paris

*Response: Done.*

Page 15, line 254: melt-down => melt

*Response: Done.*

Figure 4: Grid lines would help to be able to compare the panels. It would also be easier if they were presented in a vertical column, to emphasize the shift in time scale for higher temperatures.

*Response: We agree and have changed the figure accordingly.*

[revised manuscript text omitted]
 = \bar{h}_0 \, H/L$ being

the typical height-to-width ratio. $\bar{h}_0$ is the equilibrium-line altitude of the considered ice sheet in the initial equilibrium situation. Values for $a_0$, $\bar{h}_0$ and $L$ are later chosen to resemble the conditions of the Greenland Ice Sheet.

The non-dimensional surface elevation, h, of the ice sheet can then be expressed as

$h = \left(\frac{a}{A}\right)^{1/m}$ (5)

For the Vialov profile, m=2(n+1) where n̶ ̶d̶e̶n̶o̶t̶e̶s̶ the Glen flow-law exponent o̶b̶s̶e̶r̶v̶e̶d̶ is commonly chosen to be around n=3 which yields m=8.

We introduce the s̶u̶r̶f̶a̶c̶e̶ ̶e̶l̶e̶v̶a̶t̶i̶o̶n̶melt-elevation feedback in its simplest form through a dependency of the surface melt rate on the surface elevation:

$a = a_0 + \gamma\, \Gamma \cdot h$ (6)

with the atmospheric lapse rate $\Gamma > 0$. $\gamma$ denotes the melting sensitivity of the ice surface, i.e. the increase in surface melt-rate per degree of warming, which is regularly measured and comprises a large number of physical processes (e.g. (Box, 2013)). For simplicity we rescale the surface mass balance by the constant ice softness parameter, A̶,̶ ̶w̶h̶i̶c̶h̶ ̶i̶s̶ ̶c̶o̶n̶s̶i̶d̶e̶r̶e̶d̶ ̶t̶o̶ ̶b̶e̶ ̶c̶o̶n̶s̶t̶a̶n̶t̶.̶ 
[revised manuscript text omitted]